# A novel type of colistin resistance genes selected from random sequence space

**Michael Knopp**[1,2]*, **Arianne M. Babina**[1], **Jónína S. Gudmundsdóttir**[1], **Martin V. Douglass**[3], **M. Stephen Trent**[3,4], **Dan I. Andersson**[1]*

1 Department of Medical Biochemistry and Microbiology, Uppsala University, Sweden, 2 European Molecular Biology Laboratory, Genome Biology Unit, Heidelberg, Germany, 3 Department of Infectious Diseases, College of Veterinary Medicine, University of Georgia, Georgia, United States of America, 4 Department of Microbiology, Franklin College of Arts and Sciences, University of Georgia, Georgia, United States of America

* Michael.Knopp@embl.de (MK); Dan.Andersson@imbim.uu.se (DIA)

**Data Availability Statement:** All relevant data are within the manuscript and its Supporting information files.

## Abstract

Antibiotic resistance is a rapidly increasing medical problem that severely limits the success of antibiotic treatments, and the identification of resistance determinants is key for surveillance and control of resistance dissemination. Horizontal transfer is the dominant mechanism for spread of resistance genes between bacteria but little is known about the original emergence of resistance genes. Here, we examined experimentally if random sequences can generate novel antibiotic resistance determinants *de novo*. By utilizing highly diverse expression libraries encoding random sequences to select for open reading frames that confer resistance to the last-resort antibiotic colistin in *Escherichia coli*, six *de novo* colistin resistance conferring peptides (Dcr) were identified. The peptides act via direct interactions with the sensor kinase PmrB (also termed BasS in *E. coli*), causing an activation of the PmrAB two-component system (TCS), modification of the lipid A domain of lipopolysaccharide and subsequent colistin resistance. This kinase-activation was extended to other TCS by generation of chimeric sensor kinases. Our results demonstrate that peptides with novel activities mediated via specific peptide-protein interactions in the transmembrane domain of a sensory transducer can be selected *de novo*, suggesting that the origination of such peptides from non-coding regions is conceivable. In addition, we identified a novel class of resistance determinants for a key antibiotic that is used as a last resort treatment for several significant pathogens. The high-level resistance provided at low expression levels, absence of significant growth defects and the functionality of Dcr peptides across different genera suggest that this class of peptides could potentially evolve as *bona fide* resistance determinants *in natura*.

## Author summary

We expressed over 100 million randomly generated DNA sequences in *Escherichia coli* and selected 6 variants that encode peptides that provide resistance to the last-resort antibiotic colistin. We show that the selected peptides are auxiliary activators of the two-

**Funding:** This work was supported by grants from the Wallenberg Foundation (grant 2015.0069) and Swedish Research Council (grant 2017-01527) (to DIA) well as the NIAID grants AI129940 and AI138576 (to MST). The funders had no role in study design, data collection and analysis, decision to publish, or preparation of the manuscript.

**Competing interests:** The authors have declared that no competing interests exist.

component system PmrAB, and that resistance is mediated via modifications of the cell envelope causing decreased antibiotic uptake. This is the first example where random expression libraries have been employed to select for peptides that perform an activating function by direct peptide-protein interactions *in vivo*, adding support to the idea that non-coding DNA can serve as a substrate for *de novo* gene evolution. Additionally, the described peptides expand the narrow list of colistin resistance genes and further analyses of clinical isolates will be necessary to determine if similar resistance determinants have evolved *in natura*.

## Introduction

The increasing spread of antibiotic resistant bacteria causes hundreds of thousands of deaths per year and imposes a considerable economic burden worldwide [1]. Resistance is typically caused by chromosomal mutations, horizontal acquisition of resistance genes, or a combination of both. A rapid identification of antibiotic resistance determinants is key to predict and prevent the dissemination of multi-drug resistant isolates. Chromosomal mutations are selected during antibiotic treatment and do not spread horizontally. In contrast, antibiotic resistance genes are typically recruited on mobile genetic elements and rapidly disseminate within and between bacterial species. To understand the evolutionary origin of antibiotic resistance is of high interest for evolutionary biology as well from a medical perspective [2].

New genes mostly originate from pre-existing ancestral genes, for example by a duplication and subsequent functional divergence of the ancestral gene, or by gene fusion/fission [3]. For example, a recent study suggests that a metallo-ß-lactamase, a widespread resistant determinant with high clinical impact, likely evolved from an endoribonuclease involved in tRNA maturation [4]. While the exact evolutionary origin of antibiotic resistance genes is mostly obscure, it is widely accepted that non-pathogenic bacteria harbor a reservoir of ancestral resistance genes that spread to human pathogens due to the extensive use of antibiotics and that the substrate specificity of these genes can quickly expand by acquisition of mutations [5–7]. The transfer of novel resistance genes to human pathogens represents a major problem to modern medicine, as exemplified by the recent occurrence of the mobilized colistin resistance gene *mcr*. Colistin is a last-resort antibiotic used to treat major human pathogens that are not susceptible to other antibiotics. After the first identification of *mcr*, this class of resistance genes has been found in colistin resistant isolates worldwide, threatening the efficacy of this key antibiotic [8].

Apart from duplication-divergence and gene fusion/fission [3], new genes can also originate *de novo*, i.e. by accidental expression of normally non-functional sequences that could confer a beneficial effect. Comparative genomics has provided many convincing examples of the *de novo* emergence of genes across distant organisms [9–13], but experimental demonstration of this process is limited [14–16]. Initial studies have shown that resistance genes can be isolated from random sequences, however these *de novo* genes likely have limited clinical significance due high associated fitness costs and their inferiority to naturally occurring resistance determinants [16–18]. It remains unclear if and how *de novo* gene evolution contributes to the evolution of antibiotic resistance genes, but conceivably experimental selection of *de novo* resistance genes may allow the identification of possible resistance pathways that either have not yet emerged, or, more importantly from a medical perspective, are present but yet undetected in natural bacterial isolates.

Here, we use an experimental model system to examine if functional selections from artificially expressed random DNA sequences can be employed to identify peptides that confer resistance to the last-resort antibiotic colistin. We isolated and characterized a novel class of colistin-resistance conferring peptides that act as *bona fide* activators of the PmrAB two-component system (termed BasRS in *E. coli*) via direct interactions with the membrane sensor protein PmrB. This activation results in modification of lipid A, which is the anchor of the major surface molecule lipopolysaccharide in the outer membrane of Gram-negative bacteria, causing a reduction or redistribution of negative charge and thereby a decreased affinity to positively charged compounds including cationic antimicrobial peptides. Lipid A modifications mediated by the identified *de novo* peptides increase resistance to the last-resort antibiotic colistin, thus expanding the list of colistin-resistance determinants and identifying a new class of colistin resistance genes that could potentially emerge in nature.

## Results

### *De novo* selection of colistin resistance peptides

We generated a set of highly diverse plasmid libraries encoding randomly generated short open reading frames (sORFs) that were expressed from a strong promoter to select for *de novo* sequences that provide a beneficial effect in *Escherichia coli* [16]. The five libraries encode 10 to 50 amino acids with either no bias (rnd 10, 20, 50a), a restriction to primordial amino acids (rnd 50b) to mimic the amino acid availability supposedly present when the first genes originated *de novo* [19] or a bias for hydrophilic amino acids (rnd 50c) to promote intrinsic disorder and functional promiscuity (Fig 1A). By utilizing large scale library cloning and parallel plasmid transformations of approximately 80 pooled ligation reactions, we generated a set of over $5.8 \times 10^8$ unique sequences in total. We subjected the five libraries to a selection for variants that are able to confer resistance to the clinically important antibiotic colistin (polymyxin E) (Fig 1B). This last-resort antimicrobial peptide is classified as a 'critically important antimicrobial for humans with the highest priority' by the World Health Organization [20].

From this selection, we isolated six inserts that enabled growth of *E. coli* BW25113 at normally inhibitory colistin concentrations. The encoded peptides range from 26 to 51 amino acids and do not share sequence homologies with one another (ClustalOmega) (S1 Table) or other proteins (tblastn) using default search parameters (see Materials and methods). However, all peptides are hydrophobic and predicted to form transmembrane helices with high likelihood (Fig 1C). We termed these sORFs Dcr1-6 (*de novo colistin resistance*) and confirmed their activity by re-cloning and performing minimum inhibitory concentration (MIC) determination (S2 Table). Two variants (Dcr3,4) were isolated from the 50a library and four variants (Dcr1,2,5,6) were isolated from the 50b library. It is important to note that the hydrophilic bias for the four sequences isolated from the 50b library was lost due to frame shifts during DNA synthesis for these specific variants. We chose Dcr1 and 2 for further analyses, as these variants conferred the highest increase in resistance, increasing the MIC of colistin by 16-fold (Fig 1D). None of the peptides exhibited cross resistance to any other antibiotic class tested (Fig 1D, S2 Table). To confirm that the encoded peptide rather than the transcribed RNA was responsible for the increased resistance, we constructed two Dcr1-variants. Introduction of a stop codon in the beginning of the sORF caused a complete loss of function, while a variant that encoded the same peptide using alternative codons maintained the resistance phenotype (S1 Fig). To test whether the functionality of the peptides is restricted to certain plasmids and/or specific promoters, we made several genetic constructs: Dcr1 was also active when expressed chromosomally from an inducible $P_{LlacO}$ promoter or the constitutive J23101 and *proD* promoters, as well as in an alternative expression vector with an arabinose-inducible

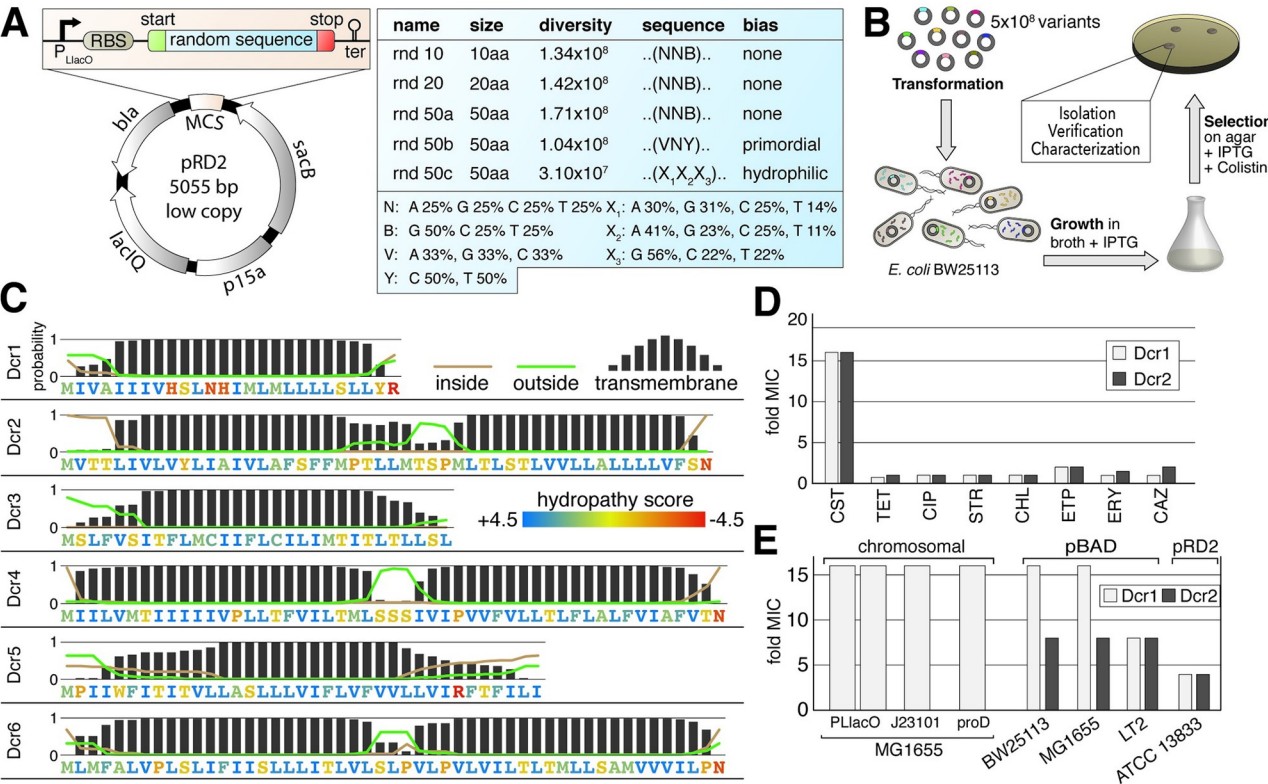

**Fig 1. Selection of peptides that confer colistin resistance.** (A) Design of the expression vector and random libraries. (B) Schematic representation of the screen for inserts conferring colistin resistance. (C) Sequence analysis of the six identified *de novo* colistin resistance peptides (Dcr 1–6). The letters represent the amino acid sequence and are color-coded for their hydropathy score. The bars and line graphs show the probability of transmembrane domains (bars) and cytoplasmic (brown) or periplasmic (green) localization. (D) Resistance profile of *E. coli* BW25113 expressing Dcr1 or 2 from the IPTG-inducible pRD2 vector compared to an empty vector control strain. Minimal inhibitory concentrations (MIC) of colistin (CST) were determined by broth dilution using Sensititre microtiter plates. MICs of tetracycline (TET), ciprofloxacin (CIP), streptomycin (STR), chloramphenicol (CHL), ertapenem (ETP), erythromycin (ERY) and ceftazidime (CAZ) were determined using Etest on agar plates. (E) MICs of colistin for various constructs expressing Dcr1 or 2 compared to empty vector control strains. All MIC determinations were performed at least in triplicate.

P_BAD promoter (Fig 1E). Notably, the chromosomal construct under the control of the P_LlacO promoter provided resistance even without induction with IPTG, which is likely due to the absence of the enhanced plasmid-encoded LacIq repressor. Since the native, less effective LacI repressor is still present in the strain used, low level basal expression during uninduced conditions appears sufficient for peptide-mediated resistance. The resistance increase was not associated with a visible detrimental growth defect, as all strains retained exponential growth rates similar to the wild type when expressing *dcr*1 or *dcr*2 from pRD2 in the presence of 1mM IPTG (S3 Table). Similarly, other growth parameters were also largely unaffected; neither the lag time (defined as time to reach OD_{600} of 0.01) or final yield (defined as OD_{600} after 16 hours of growth) decreased more than 10% compared to a wild type when expressed from pRD2 in the presence of 1mM IPTG (S2 Fig). Since antibiotic resistance is often associated with growth defects, the minor effects on fitness exerted by the Dcr peptides are considerably mild compared to other clinically relevant resistance alleles [21]. Besides *E. coli* BW25113, Dcr1 and 2 were also active in *E. coli* MG1655, *Salmonella* Typhimurium LT2 and *Klebsiella pneumoniae* ATCC 13883 (Fig 1E, S2 Table). These genera represent pathogens with major clinical relevance belonging to high (*Salmonella*) and critical priority groups (*E. coli* and *K. pneumoniae*) [22].

## Resistance is mediated by lipid A modifications

Resistance to colistin is typically achieved via cell envelope modifications that reduce affinity to colistin [23]. These modifications include the decoration of the lipid A component of lipopolysaccharide with 4-amino-4-deoxy-L-arabinose (L-Ara4N) and/or phosphoethanolamine (pEtN), which, respectively, reduce or redistribute negative charge of the cell envelope and subsequently lower affinity to positively-charged colistin (Fig 2A). Addition of these groups are catalyzed by the chromosomally encoded L-Ara4N- and pEtN-transferases ArnT and EptA, respectively, or the horizontally acquired *bona fide* pEtN-transferase *mcr*. Furthermore, PmrR is a repressor of the phosphotransferase LpxT that competes with EptA for a lipid A modification site [24] at the 1-position. Inhibition of LpxT by PmrR therefore increases pEtN modifications and reduces colistin susceptibility [24]. Expression of ArnT, EptA and PmrR is under the control of the two-component system (TCS) PmrAB, which itself is coupled to the TCS PhoPQ via the connector protein PmrD [25,26]. Apart from the plasmid-borne *mcr* gene, colistin-resistant isolates often carry mutations that constitutively activate the PmrAB pathway, causing lipid A modifications mediated via ArnT and EptA [27].

The resistance level provided by the *de novo* selected colistin resistance peptides is equal to that of *mcr* or a mutant *pmrA* allele (amino acid change G53E, hereafter denoted *pmrA**) that causes constitutive PmrAB activation (S3 Fig). Expression of Dcr1 in different resistant mutants did not further increase resistance, indicating a functional overlap. We therefore tested Dcr1 functionality in various mutants containing deletions of components of the PmrAB regulatory system (Fig 2A). PmrA, PmrB, ArnT and EptA were essential for Dcr-mediated resistance, whereas the components of the PhoPQ TCS, the connector protein PmrD, as well as PmrR and LpxT were not. Comparing the proteomic changes induced by Dcr1/2 to a *pmrA** mutant revealed an almost identical response (Fig 2B), demonstrating a full activation of the PmrAB regulon by Dcr1/2 as well as high specificity for this regulon without induction of general stress responses or other TCSs. In concordance with the genetic and proteomic analyses, the extraction and identification of lipid A species revealed a predominant decoration with L-Ara4N and pEtN moieties after Dcr1/2 expression, similar to that seen in *pmrA** mutants (Fig 2C), as the likely cause of the increase in colistin resistance [24,26].

## Dcr peptides function as auxiliary activators of PmrB

The isolated peptides act in a PmrAB-dependent manner, causing full activation of the regulon. Based on the high hydrophobicity and predictions for transmembrane helix formation, we hypothesized that the mode of action involves a direct interaction with the membrane-localized sensor kinase PmrB. While no such regulator for PmrB is known, auxiliary proteins are described in other TCSs[28–30]. For example, the functionally related TCS PhoPQ has been shown to be regulated by the membrane protein UgtL by a direct binding to the sensor kinase PhoQ [31]. The employment of a bacterial two-hybrid system, in which the proteins of interest are fused individually to two components (T18 and T25) of an adenylate cyclase, has been particularly useful for demonstrating interactions between auxiliary regulators and their corresponding kinases. Co-localization of the proteins of interest activates the two reporter-fragments, which produces cAMP and drives expression of a reporter gene (Fig 3A)[32]. We chose to test three colistin resistance peptides (Dcr1, Dcr2 and Dcr3) for interaction with PmrB using the bacterial two-hybrid system. In the initial assays, all variants lost the ability to confer colistin resistance when fused to the T25 subunit and we did not detect any interactions with PmrB. We hypothesized that the loss in function was likely due to steric hindrance from the T25 fusions. Therefore, we cloned randomized sequences (NNN repeats) between T25 and the different colistin resistance peptides to act as linkers and selected for fusion variants that

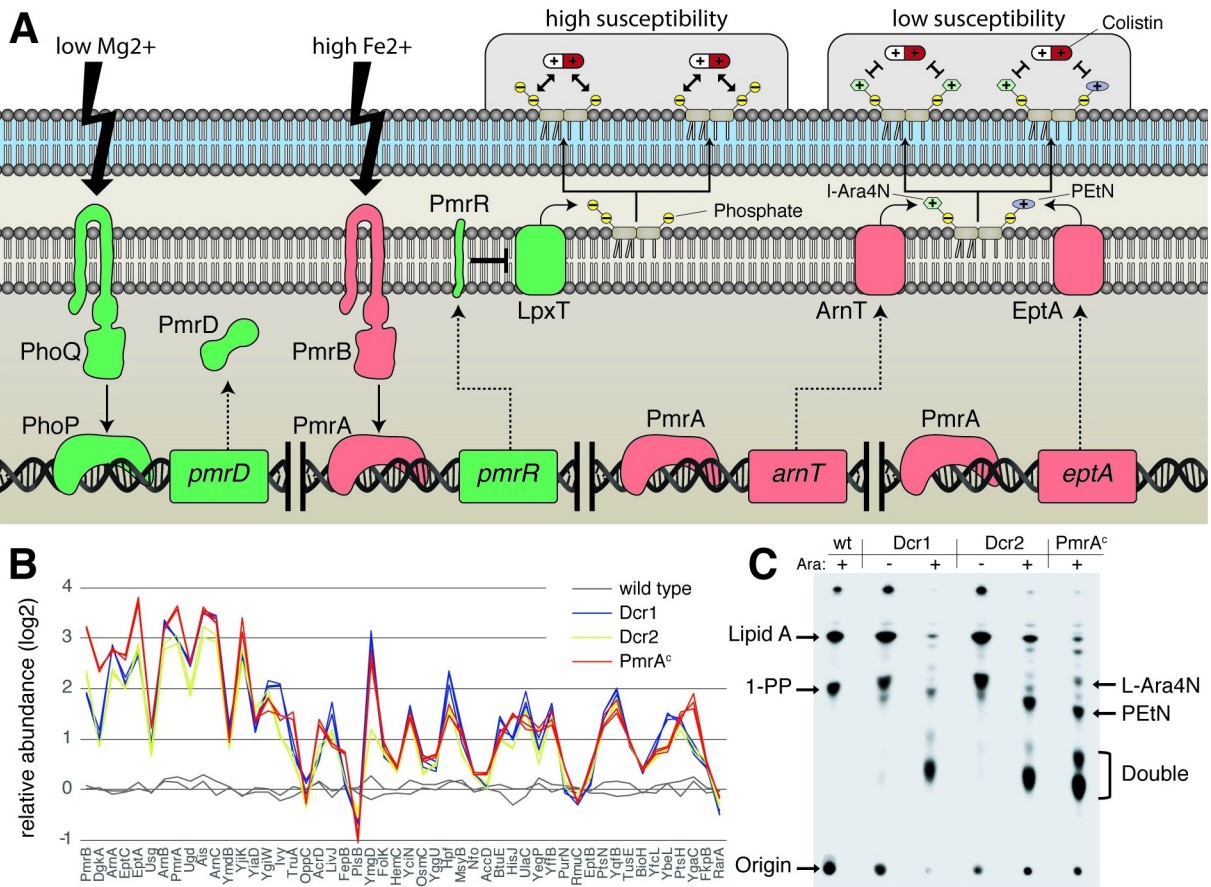

**Fig 2. PmrAB-dependent lipid A modification.** (A) Schematic representation of important proteins involved in lipid A modification in *E. coli* BW25113. The negatively-charged terminal phosphate groups increase interaction with colistin whereas the positively-charged 4-amino-4-deoxy-l-arabinose (l-Ara4N) and phosphothanolamine (pEtN) modifications decrease susceptibility. The enzymes responsible for these modifications (ArnT and EptA) are under the control of the two-component system (TCS) PmrAB, which is activated by environmental signals as well as by the TCS PhoPQ via the connector protein PmrD. LpxT, a lipid A phosphotransferase under the control of PmrR, competes with EptA for lipid A modifications. PhoPQ, PmrD, PmrR and LpxT (green) could be removed without affecting the colistin resistance provided by the Dcr peptides, while removal of PmrAB, ArnT or EptA (red) resulted in a complete loss of function of the Dcr peptides. (B) Protein abundance of cells expressing Dcr 1 or 2 and a constitutive *pmrA* mutant compared to an empty vector control strain. The top 50 proteins with the most significant change in abundance (lowest p-value) in the *pmrA* mutant are shown. Measurements of the wild type, *dcr1* and *pmrA* mutants were performed in triplicates and *dcr2* in duplicate. Abundances are relative to the wild type replicate 1. (C) TLC-based separation of lipid A species isolated from wild-type *E. coli* BW25113 with empty control vector or expressing Dcr1 or 2 with or without induction via arabinose. A *pmrA* constitutive mutant served as a positive control for lipid A modifications.

maintained functionality, i.e. the ability to confer resistance (Fig 3B). We isolated one fusion variant that contained four concatenated 10x(NNN) repeats and maintained the same level of resistance as the originally selected, non-fused Dcr3. This variant (Dcr3[L]) caused a strong activation of the two-hybrid system reporter, providing compelling evidence for an interaction between the Dcr peptides and PmrB (Fig 3C). This interaction was specific for PmrB, as no interaction between Dcr3[L] and PmrA could be detected.

The specificity observed in the two-hybrid system is in accordance with the specific upregulation of the PmrAB regulon observed in the whole proteome analysis. To test whether other TCS could be activated by Dcr, we constructed a chimeric sensor kinase based on CusSR, which is a copper- and silver-responsive TCS. We replaced either the transmembrane and periplasmic domain (Chimera-A) or the HAMP-domain (Chimera-B) of the sensor kinase

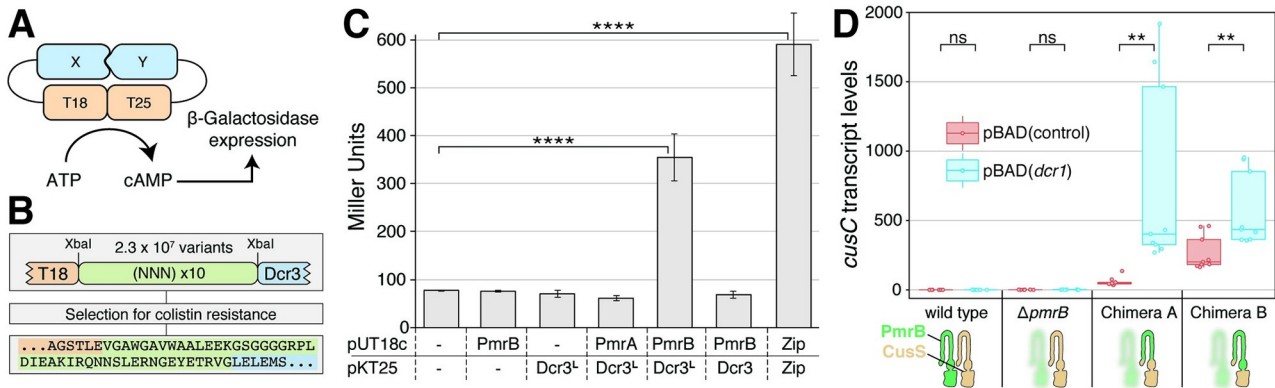

**Fig 3. Dcr peptides interact with and activate PmrB.** (A) Schematic representation of the bacterial two-hybrid system. Binding of X to Y causes co-localization of T18 and T25 and production of cyclic AMP (cAMP) that subsequently increases ß-galactosidase expression. (B) Schematic representation of the constructed linker library and the functional linker that was selected. (C) ß-galactosidase assays using various fusion constructs with T18 expressed from pUT18c and T25 expressed from pKT25. T18 or T25 without an additional fusion were used as negative controls. Zip-fusions, which contain leucine zipper motifs that are known to dimerize, were used as a positive control. pKT25-Dcr3 lost the ability to provide colistin resistance, while the variant containing the selected linker pKT25-Dcr3$^L$ regained functionality. Each value represents the mean of six independent experiments. Error bars show the standard deviation. Asterisks indicate significant differences in ß-galactosidase activity (**** = p < 0.0001, t-test). (D) Relative transcript levels of *cusC* in *E. coli* MG1655 wild type, Δ*pmrB* and two chimeric CusS variants in a Δ*pmrB* mutant expressing Dcr1 or an empty control plasmid. The chimeric sensor kinases consist of the CusS scaffold where either the transmembrane and periplasmic domain (Chimera-A) or additionally the HAMP domain (Chimera-B) were replaced with the PmrB homologous parts. The box plots show transcript levels of *cusC* determined from a minimum of three biological and three technical replicates each. The horizontal line represents the median, the box represents the upper (Q3) and lower quartile (Q1) and the vertical lines represent the highest and lowest value. Asterisks indicate significant differences in transcript level (** = p < 0.01, t-test).

CusS with the homologous PmrB domains. The native PmrB was removed to prevent possible cross-interactions. As a functional output, we measured the mRNA transcript levels of *cusC*, which encodes a component of a heavy metal efflux pump that is regulated by CusSR [33]. While no increase in *cusC* expression was detected in the wild type or *pmrB* knockout, transcript levels of *cusC* increased 8-fold when the transmembrane and periplasmic domains (Chimera A) of CusS were replaced with the homologous parts of PmrB (Fig 3D). Similarly, *cusC* expression was significantly increased when the HAMP domain was replaced. This variant also exhibited high basal expression levels even in the absence of Dcr1. This is likely due to unintended structural effects caused by the chimeric fusion of two different sensor kinases.

## Discussion

### Random sequence space can encode peptides with specific and activating functions

The origin of novel genes and functions is central to evolutionary biology and comparative genomics has produced convincing evidence of the widespread occurrence of *de novo* genes [9–13]. However, experimental validation of this process *in vivo* is scarce, mostly limited to biased libraries that rely on partial randomization in pre-defined structural scaffolds [34] and the experiments are typically designed to select for a reversion to the wild-type phenotype from a dysfunctional mutant [14–16]. Several studies have identified small peptides isolated from random or semi-random libraries to confer resistance to antimicrobial compounds [17,18], however the underlying mechanisms are largely unclear and are often associated with deleterious growth effects. In contrast, a recent study reported the selection of peptides providing a general growth benefit [35], but unfortunately the chosen set-up suffered from

experimental flaws and the results are most likely an artifact due to the inherent cost of the expression vector [36,37].

In an evolutionary context, the origination of a novel phenotype via a functional disturbance of a pre-existing mechanism is likely more easily achieved than the generation of a *bona fide* activating and novel functionality, the latter arguably being a more significant but probably rarer event in evolution. With that said, the Dcr1-6 peptides perform a specific activation, induction of the kinase activity of the sensor kinase PmrB, rather than a disruption. Interestingly, the peptides isolated from our screens, which confer aminoglycoside resistance [16] and colistin resistance (this work), are highly hydrophobic and strongly predicted to localize in the membrane. This finding concurs with recent work showing that intergenic regions are biased to encode transmembrane proteins and that hydrophobic peptides have a higher capacity for beneficial effects [38]. It is possible that the membrane presents a preferred target for *de novo* genes, as it is an integral part of almost all cellular functions and could provide a safe haven for unstructured peptides to avoid immediate degradation. Further studies utilizing libraries consisting of transmembrane domains or a strong hydrophobic bias could be useful to further elucidate the role of membrane-associated proteins and peptides in regard to the origination of *de novo* genes.

It is also important to point out the limitations of the experimental approach we used in this work and how it differs from what might occur in nature during *de novo* evolution of new genes. First, we artificially provide efficient expression signals for both transcription and translation that allow high-level expression of the random sequence library. To what extent expression of non-coding sequences is a limitation for natural *de novo* gene evolution is difficult to assess, but it is clear from recent studies that large portions of the genomes, including noncoding sequences, are highly transcribed [39] and can contain randomly occurring sORFs that are translated [40]. In addition, it has been shown that transcriptional start is easily achieved even from random sequences [41]. Therefore, expression might not be a strong constraint for *de novo* evolution since sORFs could conceivably serve as a pool for selection of functional peptides [42]. Second, the sequence library is randomized, but in a natural situation these sequences would be derived from existing sequences in the genome. This might provide constraints on exploration of sequence space, but also increase the probability of finding a new function. Thus, it is conceivable that the use of expression libraries derived from non-coding sequences in natural genomes that include, for example, pseudogenes with rudimentary sequence motifs (important for structure and/or function), could increase the potential of evolving a new function. Third, the random sequences are much shorter than a typical bacterial protein (approximately 50 versus 300 amino acids) and therefore less likely to be able to perform complex functions (e.g. enzymatic reactions). Consequently, the library has a bias towards finding peptides that confer their effect by altering the function of an existing protein/process (i.e. regulatory/structural peptides). Since 3 out of 64 codons cause translational termination, the likelihood of random sequences to encode large proteins is very low. The probability for NNN repeats to encode 50, 100 or 200 amino acids without premature termination is approximately 9%, 0.8% and 0.007%, respectively, resulting in a >1300-fold higher availability of randomly generated 50mers compared to 200mers that can serve as substrate for *de novo* gene evolution. This strong bias towards small peptides could reflect why naturally emerging *de novo* genes are typically short [42]. These *de novo* originated functional peptides could subsequently be incorporated into larger, pre-existing proteins by recombination events and gene fusion, a process which previously has been shown to promote the evolution of novel functions [43,44]. Thus, screening short open reading frames for *de novo* generated functions can be considered to be representative of *in vivo* evolution.

## Clinical relevance and implications

Resistance to colistin is rapidly increasing, reducing treatment options especially for multi-resistant Enterobacteriaceae infections. Acquired resistance to colistin is typically conferred by chromosomal mutations that modify lipopolysaccharide or increase efflux [8], but recently a mobile resistance element was identified (*mcr*)[45] and subsequently found in clinical samples globally [46]. The presence of *mcr* genes on mobile elements is particularly problematic since it allows for rapid transfer between different bacterial species [47]. In this work, we identified and characterized the mechanism of action and phenotypes conferred by a novel class of colistin resistance genes that (i) function in multiple clinically important bacterial genera, (ii) provide resistance levels equal to those of known resistance mechanisms and (iii) do not impose a severe growth defect.

So far, we have not been able to identify homologues of *dcr* in clinical isolates. However, the existence of homologues is unlikely since the six peptides described in this study lack any sequence homology at the nucleotide and amino acid levels (Fig 1A). The underlying mechanism, the modification of lipid A, is the most prevalent cause of colistin resistance and its clinical relevance is uncontested. In contrast to chromosomal mutations that cause constitutive PmrAB activation, *dcr* genes could obtain the ability to mobilize and acquire inducible regulation. This would rapidly accelerate resistance dissemination and also evade the negative effects of *pmrAB* chromosomal resistance mutations, which have an associated fitness cost [48] and a low rate of reversion to wild type physiology. A similar mobilization of mutant PmrA/B variants that are constitutively active is hampered by the presence of wild-type homologues that would still reside in the host cell even after acquisition of plasmid-borne constitutive variants. In contrast, horizontal spread of Dcr peptides circumvent that restriction by activating the native PmrAB system. Furthermore, the discovery of PmrB-activating peptides from randomized sequences suggests that PmrB-inhibiting peptides can be selected in a similar manner. Such peptides could potentially be developed into an adjunctive therapy and co-administered with colistin to increase colistin-susceptibility and hinder resistance development.

Antimicrobial peptides, such as colistin, represent an important class of antibacterial agents that pose a significantly lower potential for resistance development compared to conventional antibiotics [49,50]. However, the emergence and rapid spread of *mcr* genes providing resistance towards colistin is jeopardizing the successful use of this key antibiotic. While the clinical impact of the Dcr peptides is unknown, we provide a plausible mechanism by which resistance determinants can emerge *de novo* with competitive properties, namely a high resistance level and lack of severe growth effects, compared to evolutionary successful resistance determinants. The computational identification of Dcr homologues in clinical isolates is challenging due to the lack of sequence homology, but functional selections from genomic libraries of colistin resistant isolates that lack previously described chromosomal resistance genes (*mcr*)/mutations could be used to identify similar sORFs and provide evidence for the clinical significance of *de novo* gene evolution of antibiotic resistance genes.

# Materials and methods

## Bacterial strains and growth conditions

All strains used in this study are derivatives of *Escherichia coli* BW25113, *E. coli* K-12 MG1655, *Salmonella enterica* Typhimurium LT2 or *Klebsiella pneumoniae* ATCC13883. Unless otherwise indicated, cation adjusted Muller Hinton II (MHII) broth was used for growth in liquid medium, and supplemented with 1.5% agar for growth on solid medium. Strains carrying the pRD2 vector (GeneBank accession number MH298521.1) variants were grown in presence of

100 mg/L ampicillin for plasmid maintenance, and when appropriate with 1mM Isopropyl β-d-1-thiogalactopyranoside (IPTG) to induce expression of the encoded insert. Strains carrying pBAD18 vector [51] variants were grown in presence of 15 mg/L chloramphenicol for plasmid maintenance, and when appropriate with 0.2% L-arabinose to induce expression of the encoded insert. All strains were cryo-preserved at -80°C in MHII containing 10% dimethyl sulfoxide (DMSO). Unless otherwise indicated, all strains were grown at 37°C and liquid cultures were additionally shaken at 200 rpm.

## Library construction

The expression libraries were constructed as previously described [16]. To summarize, oligonucleotides consisting of random sequences flanked by fixed termini were synthesized (Eurofins and IDT) and complemented by primer extension. The PCR products were digested with BamHI and PstI and ligated into the in-house expression plasmid pRD2. Purification of PCR products, digestions and ligation reactions was performed using the GeneJET Gel Extraction Micro Kit (ThermoFisher) according to the manufacturer's recommendation. The purified ligation was transformed into NEB5-α electrocompetent *E. coli* (New England Biolabs) and recovered for 1.5 h in SOC medium (20 g/L tryptone, 5 g/L yeast extract, 0.5 g/L NaCl, 10 mM MgCl$_2$, 0.25 mM KCl, and 4 g/L glucose), and transformants were selected on MHII supplemented with 100 mg/L ampicillin. Cells were scraped and the plasmid libraries extracted using the NucleoBond Xtra Midi Kit (Macherey-Nagel). The libraries were subsequently transformed into BW25113 and aliquots were frozen in presence of 25% glycerol until further use.

## Selection of inserts conferring colistin resistance

Selection for colistin resistance was performed as previously described [16]. Briefly, aliquots of BW25113 carrying the expression libraries were transferred to fresh MHII broth supplemented with 100 mg/L ampicillin and 1 mM IPTG. Cultures were incubated for 3h and 200μL aliquots were spread on MHII agar containing 100 mg/L ampicillin, 1 mM IPTG and 2, 4 or 8 mg/L colistin. The plates were incubated for 24h at 37°C and colonies were re-streaked on plates with the same concentrations of colistin and IPTG. The following day, bacterial cultures were grown overnight in 3 mL MHII supplemented with 100mg/L ampicillin. An aliquot of each culture (1 mL) was cryopreserved in 10% DMSO and 2 mL subjected to plasmid extraction using the NucleoBond Xtra Midi Kit (Macherey-Nagel).

## Sequence analysis

Sequences were analysed using CLC Main Workbench v 8.1 (Qiagen). Transmembrane helix predictions were performed using the TMHMM Server v. 2.0[52]. Protein alignments were performed using Clustal Omega [53] with standard parameters (dealign input sequence, no; MBED-like clustering guide-tree, yes; MBED-like clustering iteration, yes; number of combined iterations, default(0); max guide tree iterations, default; max HMM iterations, defaults; order, aligned) and homology searches using tblastn (https://blast.ncbi.nlm.nih.gov/Blast.cgi)) [54,55] using the 'nucleotide collection' database nt (update date 2020/10/04) with default parameters (general parameters: max target sequences, 100; expect threshold, 0.05; word size, 6; max matches in a query range, 0; scoring parameters: matrix, BLOSUM62; gap cost, existence 11 extension 1; compositional adjustments, conditional compositional score matrix adjustment; filters and masking: filter, low complexity regions; mask, none). Hydropathy values were scored using ExPASy ProtParam [56] based on the amino acid hydropathicity scale by Kyte and Doolittle [57].

### Re-transformation

An overnight culture of *E. coli* BW25113 (2 mL) was washed three times with 10% glycerol and suspended in 100μL 10% glycerol. Washed cells (40μL) were mixed with 100ng of the plasmid and transferred to an electroporation cuvette (1 mm gap), electroporated using a Gene Pulser Xcell (Bio-Rad) at 1.8 kV, 400Ω and 25μF and subsequently recovered in SOC medium at 37°C with shaking at 200 rpm for 2h. Transformants were selected on MHII plates containing the appropriate antibiotic (100 mg/L ampicillin for pRD2 or pUT18c, 15 mg/L chloramphenicol for pBAD18, 12 mg/L tetracycline for pSIM5 and 50 mg/L for pKT25 transformations). No colistin was used to select the plasmid re-transformants, as this could result in the unintended selection of chromosomal mutations conferring colistin resistance.

### MIC determination

For determinations of the minimal inhibitory concentrations (MIC) using Etests (Biomerieux), cultures of the strain of interest were grown overnight in MHII supplemented with the appropriate antibiotics and diluted 1:20 in MHII supplemented with the appropriate antibiotic and inducer (1mM IPTG for pRD2 and L-0.2% arabinose for pBAD18). The diluted cell suspension was spread on MHII plates supplemented with the appropriate antibiotic and inducer using cotton swabs and an Etest strip was applied to the middle of the plate. After 18h incubation, the MIC was scored by visual inspection of the plate. For MIC determinations using microdilutions, 1–2 colonies were dissolved in 1 mL phosphate-buffered saline (PBS, 137 mM NaCl, 2.7 mM KCl, 10 mM $Na_2HPO_4$, 1.8 mM $KH_2PO_4$, pH 7.4) resulting in 0.5 McFarland. The suspension was diluted 1:100 in MHII and 50μL were mixed with 50μL MHII broth supplemented with the appropriate concentration of antibiotics to select for plasmid maintenance, inducer and a 2-fold dilution series of the antibiotic of interest in round-bottomed 96-well plates. The plates were incubated at 37°C overnight without shaking. MIC values were determined as the concentration where no growth was observed. For MIC determinations using Sensititre MIC plates (ThermoFisher), overnight cultures of the strain of interest were diluted to $10^5$ Colony forming units (CFU)/mL in MHII supplemented with the appropriate antibiotic and inducer. 50μL of the suspension were transferred to Sensititre MIC plates and incubated for 18h without shaking. MIC values were determined as describe above. All MIC determinations were performed at least in triplicate. The MIC determinations for *K. pneumoniae* strains were repeated at least 6 times, due to a higher variation across replicates.

### Growth rate determination

Overnight cultures of the strain of interest were diluted to a final concentration of $3 \times 10^6$ CFU/mL in MHII supplemented with the appropriate antibiotics and in presence of either 0.2% arabinose or 1mM IPTG to induce expression from the $P_{BAD}$ or $P_{LlacO}$ promoter, respectively. Aliquots (300μL) were transferred into Honeycomb plates and optical densities at 600 nm ($OD_{600}$) were recorded using the BioscreenC reader (Oy Growth Curves Ab Ltd) at 4 min intervals over 24h at 37°C with shaking. Relative growth rates were determined by dividing the rise of the slope during exponential phase ($OD_{600}$ 0.024 to 0.09) of the strain of interest by that of the wild type. Relative growth rates were determined using four biological replicates with two technical replicates each.

### Protein extraction for proteomic analysis

Cell pellets were homogenized using FastPrep-24 instrument (Matrix B, 0.1 mm silica beads; MP Biomedicals, OH, USA) for 4 repeated 40 seconds cycles at the speed 6.5 in 150μL of lysis

buffer containing 2% sodium dodecyl sulfate (SDS) and 50mM triethylammonium bicarbonate (TEAB). Samples were centrifuged at 13000 rpm for 10 min and the supernatants were transferred to clean tubes. The lysis tubes were washed with 100μL of the lysis buffer, centrifuged at 13000 rpm for 10 min, the supernatants were combined with the corresponding lysates from the previous step. Protein concentration in the combined lysates was determined using Pierce BCA Protein Assay Kit (Thermo Fischer Scientific, Waltham, MA) and the Benchmark Plus microplate reader (Bio-Rad Laboratories, Hercules, CA) with bovine serum albumin (BSA) solutions as standards.

## Tryptic digestion and Tandem Mass Tag (TMT) labeling

Aliquots containing 30μg of total protein were taken from each sample and reduced at 60°C for 30 min in the lysis buffer with DL-dithiothreitol (DTT) at 100mM final concentration and incubated. The reduced samples were processed using the modified filter-aided sample preparation (FASP) method [58]. In short, reduced samples were diluted to 500μL by addition of 8M urea, transferred onto Nanosep 10k Omega filters (Pall Life Sciences, Ann Arbor, MI) and washed 2 times with 200μL of 8M urea. Alkylation of the cysteine residues was performed using 10mM methyl methanethiosulfonate (MMTS) in digestion buffer (1% sodium deoxycholate (SDC), 50mM TEAB) for 30 min at room temperature and the filters were then repeatedly washed with digestion buffer. Trypsin (Pierce Trypsin Protease, MS Grade, Thermo Fisher Scientific) in digestion buffer was added in a ratio of 1:100 relative to total protein mass and the samples were incubated at 37°C overnight; another portion of trypsin (1:100) was added and the samples were incubated at 37°C for 3h. The peptides were collected by centrifugation and labelled using Tandem Mass Tag (TMT) 11plexTM reagents (Thermo Scientific) according to the manufacturer's instructions. The labeled samples were combined, pooled samples were concentrated using vacuum centrifugation and SDC was removed by acidification with 10% TFA and centrifugation.

The combined TMT-labeled samples were fractionated into 23 primary fractions by basic reversed-phase chromatography (bRP-LC) using a Dionex Ultimate 3000 UPLC system (Thermo Fischer Scientific). Peptide separations were performed using a reversed-phase XBridge BEH C18 column (3.5μm, 3.0x150mm, Waters Corporation) and a linear gradient from 3% to 40% solvent B over 13 min followed by an increase to 100% solvent B over 5 min. Here, Solvent A was 10mM ammonium formate buffer at pH 10.00 and solvent B was 90% acetonitrile, 10% 10 mM ammonium formate at pH 10.00. The primary fractions were concatenated into 10 final fractions (1+2+13, 3+4+14, . . ., 11+21, 12+22+23), evaporated and reconstituted in 15μl of 3% acetonitrile, 0.2% formic acid for nLC MS analysis.

## LC-MS/MS analysis

The fractions were analyzed on an Orbitrap Fusion Tribrid mass spectrometer interfaced with Easy-nLC1200 liquid chromatography system (both Thermo Fisher Scientific). Peptides were trapped on an Acclaim Pepmap 100 C18 trap column (100μm x 2cm, particle size 5μm, Thermo Fischer Scientific) and separated on an in-house packed analytical column (75μm x 30cm, particle size 3μm, Reprosil-Pur C18, Dr. Maisch) using a linear gradient from 5% to 35% solvent B over 75 min followed by an increase to 100% solvent B for 5 min, and 100% solvent B for 10 min at a flow of 300 nL/min. Here, Solvent A was 0.2% formic acid in water and solvent B was 80% acetonitrile, 0.2% formic acid. MS scans were performed at 120 000 resolution in m/z range 380–1380. The most abundant precursors with charges 2–7 were isolated with the m/z window 0.7 (maximum 3s "top speed" duty cycle, dynamic exclusion enabled with 10 ppm width for 60s), fragmented by collision induced dissociation (CID) at 35% energy

setting with a maximum injection time of 50ms, and the fragment spectra were recorded in the ion trap. Five most abundant MS2 fragment ions were isolated by the multinotch (simultaneous precursor selection in the m/z range 400–1400, fragmented by higher-energy collision dissociation (HCD) at 65% energy and the MS3 spectra were recorded in the Orbitrap at 50 000 resolution, m/z range 100–500.

### Proteomic data analysis

Identification and relative quantification were performed using Proteome Discoverer version 2.2 (Thermo Fisher Scientific). The reference *E. coli* K12 database was downloaded from Uniprot (December 2018) and supplemented with the mutant sequences and common proteomic contaminants (4523 sequences in total). Database matching was performed using the Mascot search engine v. 2.5.1 (Matrix Science, London, UK) with precursor tolerance of 5 ppm and fragment ion tolerance of 0.6 Da. Tryptic peptides were accepted with no missed cleavages; methionine oxidation was set as a variable modification, cysteine methylthiolation, TMT-6 on lysine and peptide N-termini were set as fixed modifications. Percolator was used for PSM validation with the strict FDR threshold of 1%.

Quantification was performed in Proteome Discoverer 2.2. TMT reporter ions were identified in the MS3 HCD spectra with 3 mmu mass tolerance, and the TMT reporter S/N abundance values for each sample were normalized within Proteome Discoverer 2.2 on the total peptide amount. Only the unique peptides were taken into account for the relative quantification on protein level.

### Isolation and analysis of lipid A species from $^{32}$P-labeled cells.

Strains were grown in LB supplemented with 100 mg/L ampicillin with 2.5 μCi/mL $^{32}$P$_i$ with either 0.2% arabinose or 0.2% glucose for inducing or inhibiting conditions, respectively. Bacteria were harvested at an $OD_{600}$ ~0.8 and washed with 5mL phosphate-buffered saline. $^{32}$P-labeled lipid A was isolated as previously described and spotted onto a silica gel TLC plate (~10,000 cpm per lane)[59]. Lipids were separated using a chloroform, pyridine, 88% formic acid, and water solvent system (50:50:16:5). TLC plates were exposed to a PhosphorImager screen and analyzed using an Amersham Typhoon Imager and software.

### Linker selection

The plasmids pKT25(Dcr1), pKT25(Dcr2) and pKT25(Dcr3) were digested with XbaI, which cleaves between the T25 and Dcr fragment. Restriction digests were purified using GeneJET Gel Extraction Micro Kit (ThermoFisher). The random linker library was prepared similarly to the random expression libraries. Here, an oligonucleotide containing 10 (NNN) repeats flanked by XbaI binding sites and a fixed primer binding site were complemented in a primer extension reaction. The resulting double-stranded product was digested with XbaI and ligated into the digested pKT25 variants at an insert:vector molar ratio of 5:1. The excess of random linker insert was chosen to increase the amount of concatenated linkers in case 10 amino acids were not sufficient to restore functionality. The ligation reactions were purified, transformed into NEB5-α cells and extracted as described above. The plasmids were then transformed into BW25113 and selected on colistin as described above. One functional linker without a stop codon was isolated (termed *dcr^L*) which was transformed into the adenylate cyclase deficient reporter strain *E. coli* BTH101[32].

## Bacterial two-hybrid assay

The bacterial two-hybrid assay kit was acquired from Euromedex and the assays were performed according to the manufacturer's recommendations with the following conditions and modifications. Strains were grown overnight from single colonies in 2 mL MHII medium containing 100 mg/L ampicillin and 50 mg/L kanamycin at 30˚C with shaking (200 rpm). These cultures were diluted 1:20 into 6 mL MHII medium supplemented with 100 mg/L ampicillin, 50 mg/L kanamycin, and 0.5 mM IPTG, and grown at 30˚C with shaking (200 rpm) until an $OD_{600}$ of approximately 0.4 was reached. Cells (1 mL) were harvested and re-suspended in 1 mL of Z buffer (50 mM $Na_2HPO_4$, 40 mM $NaH_2PO_4$, 10 mM KCl, 1 mM $MgSO_4$, 50 mM 2-mercaptoethanol [pH 7.0]). The $OD_{600}$ values for Miller Units calculations were measured as 1:2 dilutions of the cell suspensions in Z buffer. Beta-galactosidase activity assays were performed as described previously using 0.5 mL of the cell suspensions and Miller Units were calculated as follows [60]:

$$Miller\ Units = 1000 * \frac{A_{420}}{\Delta t(min) * A_{600} * vol(mL)}$$

The values reported are based on six independent biological replicates.

## Construction of CusS chimeras

In general, all chromosomal alterations were generated using the lambda Red recombineering system encoded on pSIM5-tet [61,62]. A 50 mL culture of *E. coli* MG1655 carrying pSIM5-tet was grown at 30˚C with shaking (200 rpm) in lysogeny broth (LB, 10 g/L NaCl, 10 g/L tryptone, and 5 g/L yeast extract) to an $OD_{600}$ of 0.2, moved to 42˚C with shaking for 15 minutes and quickly cooled on ice. The chilled cells were washed three times with ice-cold 10% glycerol and finally resuspended in 200μL 10% glycerol. A *cat-sacB* cassette was PCR-amplified with overhangs homologous to the chromosomal region of interest and the purified PCR product was mixed with 40μL of electrocompetent cells. The DNA/cell mixture was transformed as previously described, with the exception that the recovery of transformants was done at 30˚C to allow maintenance of pSIM5-tet for further use. Transformants were selected on LB-agar plates containing 15 mg/L chloramphenicol. To replace the *cat-sacB* cassette with the desired sequence, electrocompetent cells were prepared and transformed as described above. Recovery of transformants was done in salt-free LB overnight, to allow segregation and degradation of SacB. Transformants were selected on salt-free LB-agar plates supplemented with 5% sucrose, which is lethal to cells containing SacB. In the case of *pmrB::cat-sacB*, an oligonucleotide consisting of the two flanking sequences of *pmrB* was used during electroporation to generate a scar-free deletion. For *cusS*, the sequences of the desired chimeric constructs were designed *in silico*, synthesized by IDT Inc. and used to replace *cusS::cat-sacB*. Finally, we used the duplication-insertion recombineering method to introduce a forced duplication containing the regions of interest as previously described [63]. In brief, a *cat-sacB* cassette was used to insert a tandem duplication around the region of interest. The amplification is maintained as long as the cells are grown in presence of 15 mg/L chloramphenicol. The P1 phage was used to transduce the duplication into the target strains, which were subsequently grown on salt-free LB-agar to allow segregation of the tandem duplication and ultimately generate a strain containing the desired genetic alteration. In this manner, we first transduced the *pmrB* deletion into wild type *E. coli* MG1655. The resulting strains were used to replace *cusS* with the different chimeric variants.

## Determination of relative transcript levels

Strains were grown overnight from single colonies in 1 mL MHII medium supplemented with 15 mg/L chloramphenicol at 37˚C with shaking (200 rpm). These overnight cultures (20 uL) were used to inoculate 3 mL MHII medium containing 15 mg/L chloramphenicol and 0.2% L-arabinose, and cultures were incubated at 37˚C with shaking (200 rpm) until an $OD_{600}$ of approximately 0.5 was reached. Cells (0.5 mL) were combined with 1 mL RNAprotect Cell Reagent (Qiagen), incubated at room temperature for 10 min, and pelleted at 4˚C. Total RNA was extracted from the cell pellets using the RNeasy Mini Kit (Qiagen) according to the manufacturer's protocol. Genomic DNA was removed from approximately 5 ug of each total RNA sample using the TURBO DNA-free Kit (Invitrogen) and reverse transcription was performed using 500 ng of each DNase-treated RNA sample and the High Capacity cDNA Reverse Transcription Kit (Applied Biosystems). Quantitative PCR (qPCR) was conducted with the resulting cDNA as template using PerfeCTa SYBR Green FastMix (Quantabio) and an Illumina Eco Real-Time PCR system. Primers targeting *cusC* and housekeeping genes *hcaT* and *cysG* [64] were used to quantify transcript levels in each sample. qPCR was repeated using reactions lacking reverse transcriptase as template to confirm the effective digestion of all contaminating DNA. Data for each strain are represented by three independent biological replicates, and qPCR was performed with three technical replicates for each biological replicate.

## Supporting information

**S1 Table. Multiple sequence alignment of Dcr1-6 using Clustal Omega.** Only one site aligns as a group with strongly similar properties (indicated with ':'). No residues group with perfect ('*') or weak ('.') similarities.
(DOCX)

**S2 Table. Minimal inhibitory concentrations (MIC) of selected inserts in various backgrounds.** MICs of tetracycline (TET), ciprofloxacin (CIP), streptomycin (STR), chloramphenicol (CHL), ertapenem (ETP), erythromycin (ERY) and ceftazidime (CAZ) were determined using Etest on agar plates. ND = not determined. MICs for colistin were determined using Sensititre plates (Thermo fisher). All MIC determinations have been performed at least in triplicates.
(DOCX)

**S3 Table. Relative growth rates of *dcr1*-expressing strains compared to a wild-type control strain.** Shown are the mean of four biological and two technical replicates each. Numbers in parentheses represent the standard deviation.
(DOCX)

**S1 Fig. MICs of colistin for Dcr1 peptide variants.** Changes in the nucleotide and amino acid sequence are shown in orange.
(DOCX)

**S2 Fig. Growth characteristics of Dcr peptides.** Expression of Dcr peptides does not cause severe defects in lag time or final yield when expressed from the plasmid pRD2 (A) or from a single chromosomal copy under various promoters (B). The points represent the mean of at least three biological and two technical replicates and error bars show the standard deviation. All measurements were performed in presence of 1mM IPTG (inducer).
(DOCX)

**S3 Fig. MICs of colistin for plasmid-encoded Dcr1 and Dcr2 and the empty control vector in various genetic backgrounds of *E. coli* BW25113.** MICs for colistin were determined using

Sensititre plates (Thermo fisher). All MIC determinations have been performed at least in triplicates.

(DOCX)

## Acknowledgments

Quantitative proteomic analysis was performed at the Proteomics Core Facility, Sahlgrenska Academy, University of Gothenburg by Egor Vorontsov.

## Author Contributions

**Conceptualization:** Michael Knopp, Dan I. Andersson.

**Funding acquisition:** M. Stephen Trent, Dan I. Andersson.

**Investigation:** Michael Knopp, Arianne M. Babina, Jónína S. Gudmundsdóttir, Martin V. Douglass, M. Stephen Trent, Dan I. Andersson.

**Methodology:** Michael Knopp.

**Project administration:** Dan I. Andersson.

**Supervision:** Michael Knopp, M. Stephen Trent, Dan I. Andersson.

**Visualization:** Michael Knopp.

**Writing – original draft:** Michael Knopp, Dan I. Andersson.

**Writing – review & editing:** Michael Knopp, Arianne M. Babina, Martin V. Douglass, M. Stephen Trent, Dan I. Andersson.

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
