## [Decision Letter · Decision Letter 0]

28 Sep 2020

Dear Dr Andersson,

Thank you very much for submitting your Research Article entitled 'A novel type of colistin resistance selected from random sequence space' to PLOS Genetics. First I would like to appologize that it has taken longer than usual to send you my decision, but one of the reviewers was overloaded with work and thus needed longer to fully evaluate your manuscript. I hope this  does not cause any problems.

Your manuscript was fully evaluated at the editorial level and by independent peer reviewers. As you will see from their comments, the reviewers highly appreciated your work and the manuscript. However, each of the three reviewers also has some important points raised that need to be answered adequately in a revised version. In particular as asked for by reviewer 1, in the experimental results, the MIC values, should be from triplicate experiments. Furthermore, reviewer 2 realised that the manuscript is not in PloS Genetics style as it lacks a clear separation into introduction, results, discussion, etc. and askes for the presentation of some evidence for no cross resistance with other antibiotics which is an important point to take into consideration. The name BasRS should be used throughout the manuscript, in particular as the authors themselves mix the names as they use BasS in Fig. 2D. Finally, as listed by reviewer 3, certain data are missing and they need to be added in the revised version.

We therefore ask you to modify the manuscript according to the review recommendations before we can consider your manuscript for acceptance. Your revisions should address the specific points made by each reviewer.

[LINK]

Yours sincerely,

Carmen Buchrieser

Associate Editor

PLOS Genetics

Lotte Søgaard-Andersen

Section Editor: Prokaryotic Genetics

PLOS Genetics

Reviewer's Responses to Questions

**Comments to the Authors:**

Reviewer #1: This is an interesting paper in that it predicts a mechanism by which bacteria could evolve resistance to antimicrobials via activation of TCSs. I found the approach to be novel and accept the reported findings. I do have a few points for the authors to consider:

1. Have the authors prepared synthetic peptides corresponding to those identifed and ascertained whether they bind to PmrB in vitro? If such peptides can be made soluble, which may be an issue, would their addition to intact bacteria result in activation of PmrB?

2. In theory, the approach could be used to sensitize bacteria to cationic antimicrobial peptides via inhibiting PmrB. Perhaps the authors could discuss this possibility to assist in devloping adjunctive therapies?

3. I note that some of the experimental results, particularly MIC values, are from only duplicate experiments. For the sake of rigor, results from triplicate experiments should be reported as mode values.

Reviewer #2: The authors describe their work to identify and characterise novel peptides that, when expressed in E. coli, lead to resistance to the antibiotic colistin. The work is performed to a high standard and the conclusions are supported by the data. The experiments to determine that one of the identified ‘synthetic’ colistin resistance peptides acts through directly interacting with the two-component sensor histidine kinase BasS/PmrB are particularly elegant.

Throughout the manuscript the authors discuss the two-component system PmrAB, but this system is termed BasRS in E. coli (Nagasawa et al. J Biochem. 1993 114(3):350-7.) while the name PmrAB is used in other Gram-negatives (Salmonella, Klebsiella). I would recommend the authors to use the term BasRS throughout the manuscript, or, at a minimum, indicate that BasRS is a synonym for this two-component system in E. coli. Indeed, I notice that the authors use BasS in Fig. 2D.

In addition, there are data lacking in the manuscript (and these need to be added), and some typos that need fixing. These are further outlined below.

L. 55: correct ‘pear’ to ‘per’

L. 100: within the context of this study, it is not entirely obvious why the authors have chosen to make a library based on amino acids that have been present on the primordial Earth. This should be explained in more detail. Perhaps more importantly, I was unable to find information on the libraries from which Dcr1-6 were selected and this information should be added to the manuscript.

L. 116 - 120: for clarity, the rationale for generating the chromosomal construct may be described first, before the results are discussed.

L. 130: the data for Klebsiella and Salmonella are currently not included in the manuscript but should be provided.

L. 159 - 178: this section is confusing as it discusses Dcr-3, but the data in Fig. 3 appear to show data for another Dcr (Dcr-1). This should be checked and corrected. Similar to my comment on Fig 3 (see below), the heading of this section should be changed as data for only one Dcr are provided. Indeed, the data discussed in L.168-169 need to be provided in the manuscript.

L. 192: please add the heading ‘Discussion’ here.

L. 297: correct ‘over night’ to ‘overnight’

L. 298: complete line after 100 mg/L.

L. 314: ‘appropriate antibiotic’ is perhaps somewhat ambiguous. Can the authors be more specific and confirm that they did not include colistin in the plates on which they selected transformants?

Fig. 3. The title of the figure should be changed to reflect that this figure does not show data of multiple peptides identified in this study, but only the Dcr-1 peptides (see also comment above). In panel (B), correct ‘vaiants’ to ‘variants’

Reviewer #3: This is an interesting manuscript describing the results of an experiment where over 100 million randomly generated DNA sequences in Escherichia coli were tested for resistance to colistin. This resulted in six variants that encoded peptides providing resistance. It is shown that these peptides are activators of a two-component system resulting in decreased antibiotic uptake. This is interesting for the topic of antibiotic resistance (colistin being a last resort antibiotic) and for the topic of evolution of novel functions (since the function emerges from random sequences). This goes in the line of previous ideas that non-coding DNA can serve as a substrate for de novo gene evolution. The findings are very interesting and worthy of publication. The paper is clear and well-written.

I first gave some consideration on whether this is novel enough for PlosG. The approach was published before by the authors in a mBIO paper focusing on aminoglycosides where it was also found that small peptides affect membrane structure and thus antibiotic uptake. I do think the present manuscript is novel enough because it focuses on another antibiotic, the peptides are not the same and the discussion around the evolutionary consequences of the results is much more developed here than in the previous publication.

The methods detailing the computational analyses are not precise enough. One needs proper citations of the methods, indication of version numbers, and of the parameters that were used to run the program and filter the significant results (where relevant).

Line 112. I would have expected the presentation of some evidence for no cross resistance with other antibiotics. It's an important point.

The discussion on small random genes being possible in genomes is interesting. It might merit some complement, since recombination/fusion can incorporate these genes into larger ones and thus provide novel functional domains to existing proteins.

The manuscript lacks a clear separation into introduction, results, discussion, etc. To the best of my knowledge this does not fit the format of PlosG.

**Have all data underlying the figures and results presented in the manuscript been provided?**

Reviewer #1: Yes

Reviewer #2: **No: **See comments to authors.

Reviewer #3: Yes

PLOS authors have the option to publish the peer review history of their article (what does this mean?). If published, this will include your full peer review and any attached files.

Reviewer #1: No

Reviewer #2: **Yes: **Willem van Schaik

Reviewer #3: No

---

## [Editor Report · Decision Letter 1]

27 Oct 2020

Dear Dr Andersson,

We are pleased to inform you that your manuscript entitled "A novel type of colistin resistance selected from random sequence space" has been editorially accepted for publication in PLOS Genetics. Congratulations!

Yours sincerely,

Carmen Buchrieser

Associate Editor

PLOS Genetics

Lotte Søgaard-Andersen

Section Editor: Prokaryotic Genetics

PLOS Genetics

Comments from the reviewers (if applicable):

**Data Deposition**

http://datadryad.org/submit?journalID=pgenetics&manu=PGENETICS-D-20-01241R1

**Press Queries**

---

## [Editor Report · Acceptance letter]

18 Nov 2020

PGENETICS-D-20-01241R1 

A novel type of colistin resistance genes selected from random sequence space 

Dear Dr Andersson, 

We are pleased to inform you that your manuscript entitled "A novel type of colistin resistance genes selected from random sequence space" has been formally accepted for publication in PLOS Genetics! Your manuscript is now with our production department and you will be notified of the publication date in due course.

With kind regards,

Nicola Davies

PLOS Genetics

On behalf of:
